# Breast Reduction with Implants or Augmentation Reduction: Patient-Reported Outcomes from a Single-Centre Retrospective Cohort Analysis

**DOI:** 10.3390/medicina60050743

**Published:** 2024-04-29

**Authors:** Derek Liang, Vicky Edwards, Giuseppe Di Taranto

**Affiliations:** 1Chelsea and Westminster Hospital NHS Foundation Trust, 369 Fulham Rd., London SW10 9NH, UK; derek.liang@nhs.net (D.L.); vedwards@enhancemed.co.uk (V.E.); 2Hammersmith Private Hospital, Enhance Medical Group, The Triangle, 5-17 Hammersmith Grove, London W6 0LG, UK

**Keywords:** breast reduction with implants, augmentation reduction, breast augmentation

## Abstract

*Background and Objectives*: The correction of breast hypertrophy and ptosis with implant placement has always posed a challenge for plastic surgeons. Various methods have been devised, yielding conflicting results. The purpose of this study is to describe our surgical technique of breast reduction with silicone implants, present the safety profile of the procedure, and report patient-reported outcomes. *Materials and Methods*: A retrospective review was performed on our case series of cosmetic breast surgery performed by the senior author between October 2020 and November 2023. Only patients who had over 300 g of breast tissue removed were included. The surgery and demographic characteristics were recorded. Patients were asked to complete a questionnaire about satisfaction with their breasts pre-operatively and after the surgery. *Results*: Over 745 cases were performed, and 25 were included in the analysis. In total, 78.3% of the patients presented with a Grade 3 ptosis. The mean implant size was 352.39 cc (range 300–455 cc). The breast tissue removed ranged from 312 to 657 g. The mean follow-up was 14.17 months. Only one case required revision surgery after developing capsular contracture and a waterfall deformity. Patients reported a statistically significant improvement across all domains of the questionnaire (*p* < 0.001). *Conclusions*: Breast reduction plus implants is a safe and effective alternative for patients with large ptotic breasts who wish to attain a full upper pole. It carries a similar risk profile to augmentation mastopexy and maintains its functional benefits in alleviating back, neck, and shoulder pain.

## 1. Introduction

Breast reduction surgery has historically been focused on function, with cosmesis as a secondary surgical outcome. Over the decades, techniques have been developed and refined to improve the shape, offer maximal tissue resection, and minimize complications in an attempt to provide reliable long-term results [1,2,3,4,5]. Despite these refinements, these traditional methods of breast reduction surgery do not typically achieve long-term upper pole fullness or convexity [6,7]. In patients who have significant breast volume and ptosis, combining breast reduction with implant augmentation can be an effective one-stage procedure to simultaneously correct ptosis and create a round upper pole. The idea of replacing already resected breast parenchyma with an implant and then subjecting the overlying skin to high-stretch forces may seem counterintuitive. One might assume that these patients would be at risk of a high rate of complications; however, this is not the case. Although there are few, current studies [6,8,9,10,11] have shown that the complication profile of breast reduction plus implants is similar to those of mastopexy augmentation. In light of the limited pre-existing research, we conducted a retrospective cohort analysis of patients who received breast reduction plus implants by the senior author (GDT). This study aims to highlight the outcomes and complication profiles in patients receiving breast reduction plus implants, along with presenting patient-reported outcome measures for the procedure. We will also compare this to the existing literature in order to support and instill confidence in practitioners wanting to advance their plastic surgical armamentarium.

## 2. Materials and Methods

A thorough review was performed of all cosmetic breast cases performed by the senior author in the UK between October 2020 and November 2023. The data of all the surgeries are prospectively collected within the online database of the hospital. Only patients who had over 300 g of breast tissue removed per breast and a follow-up longer than 3 months were included in this study group. Patients and surgery characteristics were recorded. Patients were also asked to complete a pre-operative questionnaire pertaining to their satisfaction with their breasts on a 4-point Likert scale. The same questionnaire was also repeated post-operatively at 3 months. This was recorded for patient-reported outcome measures (PROMs). The difference in the pre-operative and post-operative scores was analyzed.

### 2.1. Surgical Technique

All patients were marked pre-operatively standing up with standard landmarks (midline, breast meridian, inframammary fold, and upper breast pole). The new position of the nipple apex was also marked 2 cm above the projection of the inframammary fold to the anterior skin.

Cases were performed under general anesthesia with the patient supine and their arms by the side. The breast implant was placed first via a horizontal skin incision at the inframammary fold. Dual-plane dissection was performed to allow the implant to sit in the subpectoral space. We exclusively used Mentor textured moderate- or high-profile implants. The superficial fascia was then reapproximated to provide coverage of the implant. Parenchymal resection was then performed. All breast reductions were performed with a superior pedicle and Wise skin pattern resection [12]. Breast parenchyma was excised in a conical fashion, starting from the centre of the lower pole and ensuring the superficial fascial system covering the implant was not breached (Figure 1). This resulted in two flaps of breast tissue (medial and lateral), which were then sutured together with a continuous suture to recreate a smooth rounded lower pole and increased projection of the breast. This avoided an empty or boxed shape, which typically occurs in a traditional breast reduction [13]. Symmetry of the nipple was checked at the end of the procedure with the patient sitting up. No drains were used for the operation.

All patients were discharged within 24 h and advised to commence using a surgical bra for 6 weeks. The first outpatient follow-up was performed at 7 days post-operatively, with subsequent reviews as required until all wounds were healed. All patients had a 3-month follow-up for surgical review, and post-operative PROMs were collected at this point in time. Patients were permitted to return to work after 14 days post-operation, but physical exercise was only allowed after 4 weeks.

### 2.2. Statistical Analysis

Data were collected as a retrospective chart review and included age, body mass index (BMI), smoking status, sternal notch-to-nipple distance, ptosis grade, implant size, weight removed from each breast, and PROM findings.

Statistical analyses were performed using IBM SPSS (SPSS, IBM Corp., Armonk, NY, USA). Descriptive statistics included mean and standard deviation (SD) for continuous variables and percentages for categorical variables.

Responses on the 4-point Likert scale questionnaire were converted numerically (1 = very dissatisfied, 2 = somewhat dissatisfied, 3 = somewhat satisfied, and 4 = very satisfied) and tested for significance using paired t-tests. Statistical significance was achieved if the alpha was less than 0.05.

## 3. Results

A total of 745 cosmetic breast cases were performed. Twenty-five cases were breast reductions with implants/augmentation. Only 23 of those patients fitted the inclusion criteria. All surgeries were performed at Hammersmith Private Hospital, London, United Kingdom. The median age was 29 years (range 19–40 years). The median BMI was 24.9 (range 22.2–32.5). Just under 35% of patients (8 out of 23) were smokers. In terms of breast measurements, the average sternal notch-to-nipple distance was 28.43 cm and 28.67 cm for the right and left breasts, respectively. Grade 3 breast ptosis was the most common finding (78.3%) in our cohort; the rest were grade 2. The average implant size was 352.39 cc (range 300–455 cc). The mean amount of breast tissue removed from the right breast was 391.78 g (range 312–657 g), and the mean amount of tissue removed from the left breast was 392 g (range 340–684 g). The mean follow-up was 14.17 months, ranging from 3 to 26 months (Table 1). There were two complications in total: one patient developed a capsular contracture and a waterfall deformity at 1-year post-operation, which required a corrective operation. The other patient had a correction of an inverted nipple at the same time as the breast reduction with implants and developed an infection of the nipple-areolar complex, which was managed effectively with oral antibiotics.

In regard to the questionnaire findings, pre-operatively, there were no patients who reported being “very satisfied” when asked, “How satisfied do you look in the mirror clothed?” However, there was a relatively equal distribution in the other degrees of satisfaction. Satisfaction rose significantly post-operatively, with 17 (73.9%) patients reporting being “very satisfied” when asked the same question. There were no patients who were “very dissatisfied”, and there was only one patient who was “somewhat satisfied”. This result is statistically significant (*p* < 0.001). When asked pre-operatively, “How satisfied that your breast size matches the rest of your body?”, only one patient was “very satisfied”, with ten patients “somewhat satisfied”. Post-operatively, 21 (91.3%) patients reported being “very satisfied” with how their breast size matches with the rest of their body. Only one patient expressed dissatisfaction. This is also statistically significant (*p* < 0.001). When asked, “How satisfied your bras fit?” pre-operatively, there were no patients who reported being “very satisfied”. The majority of patients (73.9%) expressed some form of dissatisfaction. In contrast, 20 patients (86.9%) were “very satisfied”, and 2 patients were “somewhat satisfied” post-operatively, which resulted in a satisfaction rate of 95.7% (*p* < 0.001). When asked, “How satisfied are you with the cleavage you have when you wear a bra”, seven patients reported being “very dissatisfied”, and eight patients reported being “somewhat dissatisfied” at the pre-operative stage. When asked the same question post-operatively, no patients were dissatisfied. In total, 19 patients reported being “very satisfied”, and 4 patients were “somewhat satisfied”, indicating a 100% satisfaction rate (*p* < 0.001). Eleven patients were “very dissatisfied”, and nine patients were “somewhat dissatisfied” when asked pre-operatively, “How satisfied are you with the size of your breasts?”. When converted to a numerical Likert value, this was represented as a mean of 1.65, which lies in between “very dissatisfied” and “somewhat dissatisfied”. No patients were “very satisfied”, and only three were “somewhat satisfied”. Post-operatively, 19 patients were “very satisfied”, and 4 were “somewhat satisfied” with the size of their breasts, with a mean value of 3.82 as a numerical representation of the Likert scale. No patient expressed any form of dissatisfaction, and this was found to be statistically significant (*p* < 0.001). Out of all the questions in the questionnaire, patients reported being “very dissatisfied” the most (19 out of 23 patients) when asked pre-operatively, “How satisfied are you with how you look in the mirror unclothed?” This resulted in the lowest mean numerical Likert score (1.30). Post-operatively, this figure increased to 3.61, with 18 (78.3%) patients now reporting being “very satisfied” with how they look unclothed. Only three patients described some form of dissatisfaction. This marked improvement on the numerical Likert scale was statistically significant (*p* < 0.001).

When comparing the findings of the pre-operative and post-operative questionnaires, there is a statistically significant improvement (*p* < 0.001) across all parameters on the Likert scale (Table 2) (Figure 2 and Figure 3).

## 4. Discussion

For patients undergoing a traditional breast reduction, it is difficult to achieve long-lasting upper pole projection. Modifications, such as moving away from an inferior pedicle and adopting a vertical skin pattern resection, are known to improve upper pole fullness [13]. Despite this, we have found in our practice that there is still a subset of patients who desire a more marked cleavage that is unattainable with traditional techniques. Although it may seem counterintuitive, breast reduction followed by implant insertion allows the large ptotic breast to be simultaneously reshaped and lifted and appear fuller at the upper pole. Studies by Chasan [10], Guimaraes [11], Sakai [8], Manero [9], and Swanson [6] all describe the principle of breast reduction followed by implant augmentation and its benefits for achieving superior upper pole fullness and breast projection. However, the nomenclature is varied across all the studies. Terms such as “Reductive augmentation”, “Structured mammaplasty”, “Combined breast reduction augmentation”, and “Breast reduction with implants” all refer to the principle of performing large resections of the breast parenchyma, followed by the insertion of implants. We agree with Swanson 6 that the term “breast reduction plus implants” avoids confusion and should be the standard when referring to this type of procedure.

So far, there has only been a limited number of studies exploring the concept of breast reduction plus implants. To our knowledge, there have only been five studies [6,8,9,10,11] that describe the idea of resecting moderate amounts of breast parenchyma and placing implants at the same stage with the aim of increasing upper pole fullness. Even within the aforementioned literature, there are interstudy variables that make it difficult to define exactly what constitutes breast reduction plus implants. Variations include the type of skin and method of parenchymal resection, the location of implant placement, and, most importantly, the amount of tissue resected. Chasan [10] presents in his study a series of 35 patients who underwent the resection of breast parenchyma with concomitant implant placement, which he named reductive augmentation. Twelve of these patients had primary macromastia and desired superior pole fullness and more anterior projection. The second subset of 23 patients were patients who previously had breast augmentation but had now developed excessive pseudoptosis and breast asymmetry. For those patients receiving primary breast reduction plus implants, his surgical technique involves initially performing an infra-areolar vertical skin incision to access the subpectoral space. The dual-plane technique is utilized for implant placement. Tailor tacking is performed next, with the patient sitting up to determine the skin resection. This is followed by breast parenchyma resection using a superomedial pedicle technique. The average tissue removal in their primary group was 255 g (range 55–465 g) per breast, and the average tissue removal in the revision group was 227 g (range 55–570 g). Guimaraes et al. 11 call their technique reduction mammaplasty with the use of implants. Their study is a case series of 15 patients undergoing their procedure. They have described their surgical technique as the tower mammaplasty, involving the excision of skin and lower pole breast parenchyma via an infra-areolar vertical incision approach. The superior limit of the skin excision is 1 cm below the inferior border of the areolar, and the inferior limit is the inframammary fold. A point is marked on the inframammary fold 9 cm lateral to the body’s midline. Next, 2 cm of skin is taken from either side of this point, marking the medial and lateral extents of the skin excision. Thus, the width of the excision is 4 cm. The lower breast parenchyma is excised with skin in the shape of a keel, extending perpendicularly to the pectoral fascia. The implant is placed in a subglandular plane. The average weight of resected breast tissue was 165 g (range 75–255 g). Sakai et al. 8 present a case series of 100 patients, 68 of which were primary operations and 32 secondary surgeries. They term their technique as structured mammoplasty. In their primary operations, the lower pole breast parenchyma is resected via a Wise skin pattern. This lower pole resection only includes skin and subcutaneous tissue. The remaining breast parenchyma is completely dissected off the pectoralis major, forming medial and lateral pillars and a superior nipple-areolar complex pedicle. A wide protective lower pole flap of superficial fascia is preserved to provide lower pole coverage and support for the eventual implant. Subpectoral dissection is then performed to allow for subpectoral placement of the implant. Once the implant is placed, the lower border of the pectoralis major and the thin fascial flap are sutured together to provide complete coverage of the implant. Finally, pillar sutures, nipple repositioning, and skin closure are performed. The average weight of resected breast tissue per breast was 340 g (range 150–620 g). Manero et al. [9] detail a cohort of 366 patients who receive some form of breast parenchymal resection followed by implant placement. They define their technique as a combined breast reduction augmentation. Interestingly, they separated their cohort into two groups based on the size of the resection. Patients were deemed to have had a breast reduction augmentation if the weight of each breast weighed more than 200 g; otherwise, they were categorized into the mastopexy group. The surgical technique did not differ between both groups. Resection of the breast parenchyma was performed via a Wise pattern skin resection. The superomedial pedicle was standard; however, some patients in their group received nipple grafts if the nipple-areolar complex transposition exceeded 8–10 cm. Superior, medial, and lateral breast flaps were completely elevated off the pectoralis major fascia. Parenchymal resection was further achieved by thinning these flaps to 2 cm in thickness. Subpectoral dissection was performed to allow implant placement, and this dissection was extended to the serratus and rectus abdominis fascia if required in order to allow complete submuscular coverage of the eventual implant. Out of the 366 patients, 182 were defined as having breast reduction augmentations. The average weight of breast tissue resected was 520 g per breast (range 202–2308 g). Swanson 6 presents a comparative study between breast reduction and breast reduction plus implants within his own cohort of 80 patients. This surgical technique involved vertical skin pattern resection with a medial pedicle. Implants were placed in the subpectoral space. In total, 26 patients in this study received breast reduction plus implants. The average weight resected for each breast was 370 g (range 129–680 g) for the right breast and 368 g (range 195–724 g) for the left.

From these studies, it is clear that the concept of breast reduction plus implants is nuanced and has a wide variation in terms of surgical technique, nomenclature, and definition. This is unsurprising given the multitude of surgical approaches to breast reduction [1,2,3,4,5] and breast augmentation alone [14,15,16]. The commonality, however, between all studies is the intent of removing as much breast parenchyma as possible to correct pseudoptosis and utilizing the breast implant to fill and expand the upper pole of the breast. The notion of size being a deterministic marker for whether a procedure is deemed a breast reduction plus implants versus mastopexy augmentation is confusing. In the five studies already published, there seems to be no consensus. Swanson [6] classifies this as a resection of greater than 300 g, Manero [9] lowers this to 200 g, whereas Chasan [10], Guimaraes [11], and Sakai [8] do not make that distinction. In our study, we make the distinction of greater than 300 g resection, following Swanson’s 6 classification. However, it must be noted that Swanson’s reason for setting the 300 g limit is so that he could make comparisons with his other studies for which he had made the same cut-off [17,18,19]. As far as we know, there is no scientific basis for a weight cut-off; however, it could be a result of fulfilling arbitrary insurance criteria in order for patients to qualify for coverage [20]. It may be simpler to define breast reduction plus implants as a procedure that intends to primarily remove excess breast parenchyma in a patient that would have otherwise received a standard breast reduction with the option of implants, allowing these patients to achieve a fuller appearance that was not previously available. The problem with this is that clinically, there may be no difference between breast reduction plus implants and augmentation mastopexy, given that the surgical technique remains very similar, and breast resection weights may overlap. For now, until more research on breast reduction plus implants comes to light, we can view these two operations as sitting within a spectrum with some areas of overlap in between.

Clinical concerns regarding breast reduction plus implants include the following:An increase in complication profile with large breast parenchyma resection coupled with significant skin assembly along with implant insertion contributing to high skin tension;Nullification of the functional benefits provided by a breast reduction, as the resected volume is replaced with an implant.

In our study, we report our outcomes in 23 patients who sought a breast reduction and desired breast implants to attain more breast projection and a fuller upper pole. The complication rate in our cohort was 8.7% (one breast-related complication and one implant-associated complication), which is comparable to the reported literature at 6.58–50% [6]. Only one patient (4.3%) required reoperation for a capsular contracture and waterfall deformity; this, too, is comparable to the rate of reoperation in the reported literature of 08–20% 10. When compared to studies addressing single-stage augmentation mastopexy, the complication rates are also similar at 13.1–32.9% [21,22,23]. Within our practice, the senior author has performed 122 single-stage augmentation mastopexies, with a complication rate of 3.3% (two waterfall deformities requiring revision, one capsular contracture, and one infection). Although this complication rate is less than the 8.7% found in our cohort of breast reduction plus implants, it is in keeping with the reported rates within the literature [14,15,16], which has wide variability. Accounting for a potential early learning curve period for the breast reduction plus implant cohort, the expectation is that the complication rate may reduce and start to resemble that of the augmentation mastopexy group. Based on the findings within our practice, in our hands, breast reduction with implants, generally, has a similar and comparable complication profile to single-stage augmentation mastopexy.

It Is known that the amount of parenchymal resection is not the only factor in alleviating symptoms of macromastia (back, neck, and shoulder pain, intertrigo, and reduced exercise tolerance). Arguably, just as important for symptomatic control is the rearrangement of the breast parenchyma and addressing the ptotic breast [24]. Therefore, even though implant insertion negates the weight loss in each breast after reductive surgery, it is specifically the excision of the inferior pole tissue and breast lift that improves the functional outcomes for these patients. This is highlighted by Swanson 6, who demonstrated that all patients who received breast reduction plus implants no longer had problems with exercise post-operatively (81.3% of patients had poor exercise tolerance pre-operatively). This discredits the notion that implant insertion at the time of breast reduction hinders its functional benefits. Although functional issues, such as back, shoulder, and neck pain, are not addressed in our PROMs, within our patient cohort, we have not found complaints or a desire to remove the breast implants due to worsening functional outcomes.

The findings from our pre-operative and post-operative questionnaires show that patients, overall, reported a statistically significant improvement in satisfaction across all six question parameters (Table 2). This supports the effectiveness of breast reduction plus implants in attaining an outcome desired by patients (Figure 1 and Figure 2).

There have been other techniques described to improve superior pole fullness, such as auto-augmentation with dermal suspension [25], parenchymal flaps [26,27], chest wall-based flaps [28,29], and mesh [30,31]. However, long-term results with these techniques have been unreliable [32]. The most popular type of auto-augmentation is the inferiorly based parenchymal flap [26,27]; however, due to the inferior base, it is subject to ptosis over time, similar to an inferiorly based pedicle in breast reductions [33]. Chest wall-based flaps [28,29] have also been described in an attempt to recreate upper pole fullness by passing breast tissue underneath a loop of pectoralis major. The issue with this is that the amount of breast tissue able to be recruited in this case is variable and, in most cases, is inadequate to completely fill the upper pole compared to implants [10]. Furthermore, the flaps are inferiorly based, which again subjects them to bottoming out over time. Meshes can be permanent or partially absorbable with materials such as vicryl or mixed vicryl. It supposedly provides a strong and robust support for the underlying breast parenchyma, which, in turn, prevents ptosis. Theoretically, the introduction of a mesh can generate a degree of foreign body reaction and scarring, which may yield unfavorable cosmetic results 9. Currently, early research shows that meshes provide protection against ptosis, but the long-term results in the future prove differently [34].

There are some limitations to this study, primarily its retrospective nature and small sample size (23 patients). This could potentially subject the study to selection bias, with certain associations, such as BMI to complications/questionnaires, unable to be made due to underpowering. There was a varied follow-up time across the patient cohort due to the difficulties in obtaining long-term follow-ups in our cosmetic patient population group. As such, long-term complications could be misrepresented and not completely captured in this patient cohort. As a result of the wide range of follow-ups in our patient population, we found that sending questionnaires at 3 months allowed us to capture the majority of our cosmetic patient population. However, we acknowledge that measuring patient satisfaction at 3 months post-operatively may not be reliable in gauging long-term overall cosmesis. Since the practice’s inception, the 6-question pre- and post-operative questionnaires have been used for all our cosmetic patients. This was introduced in order to satisfy accreditation standards from the Care Quality Commission in the United Kingdom. We realize that our questionnaire is similar but not exactly the same as the validated tool, BREAST-Q. This is a further limitation of the study; ideally, we would employ BREAST-Q as our PROM tool. In fact, our practice is currently in the process of switching all of our questionnaires to BREAST-Q.

## 5. Conclusions

Breast reduction plus implants is a safe and effective alternative for patients with large ptotic breasts who wish to attain a full upper pole. In our practice, it carries a similar risk profile to augmentation mastopexy and does not worsen functional impediments, such as back, neck, and shoulder pain, which are normally found in patients with macromastia. Patients report a high level of satisfaction post-operatively, which indicates the operation effectively addresses their pre-operative desires. With this information, we hope that breast reduction with implants will not be feared or questioned but approached with enthusiasm and a willingness to learn.

## Figures and Tables

**Figure 1 medicina-60-00743-f001:**
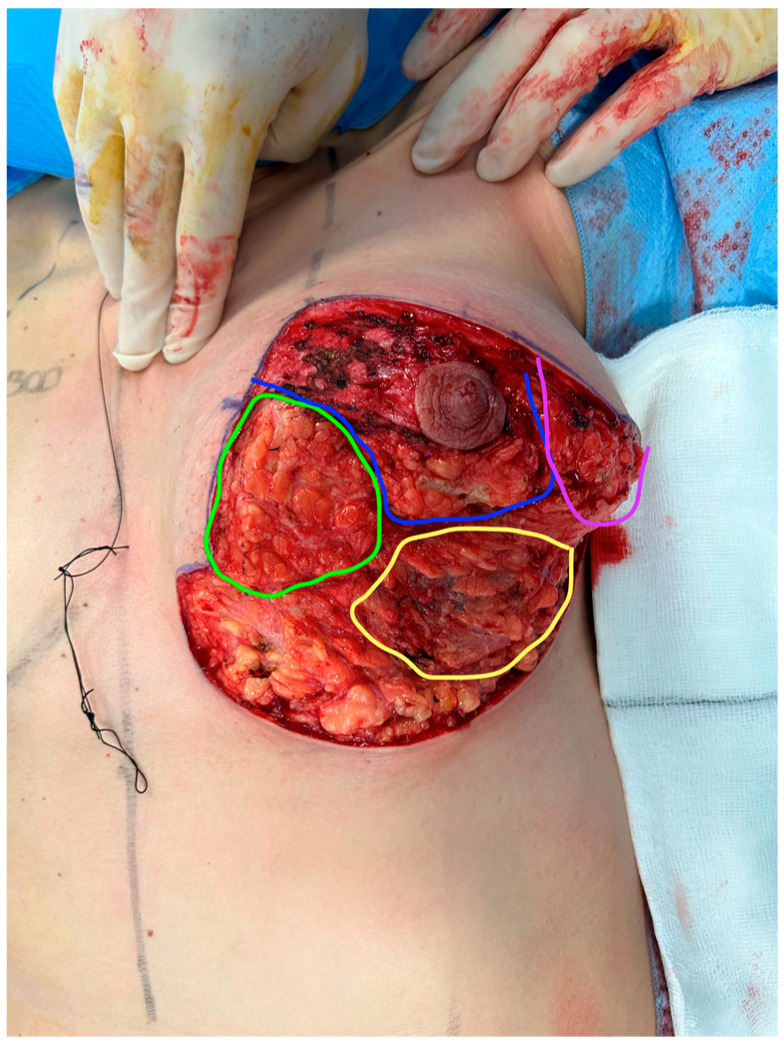
Intraoperative photo of left breast after implant placement and parenchymal resection. Green—medial pillar flap, purple—lateral pillar flap, blue—superior nipple areolar pedicle, yellow—area of closed superficial fascia and pectoralis major with implant sitting underneath.

**Figure 2 medicina-60-00743-f002:**
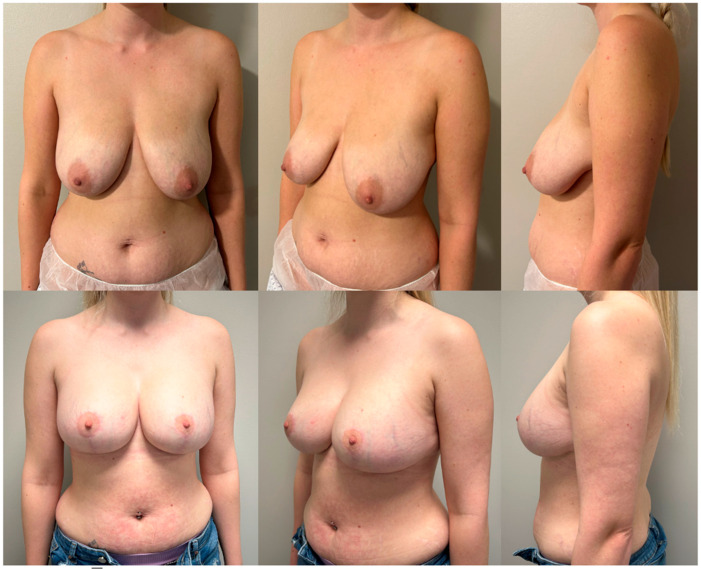
Pre-operative pictures (**above**) and post-operative pictures (**below**) of a 31-year-old patient with a grade 3 ptosis who underwent breast reduction and augmentation with 350 cc round moderate plus profile implants. Results at 1-year follow-up.

**Figure 3 medicina-60-00743-f003:**
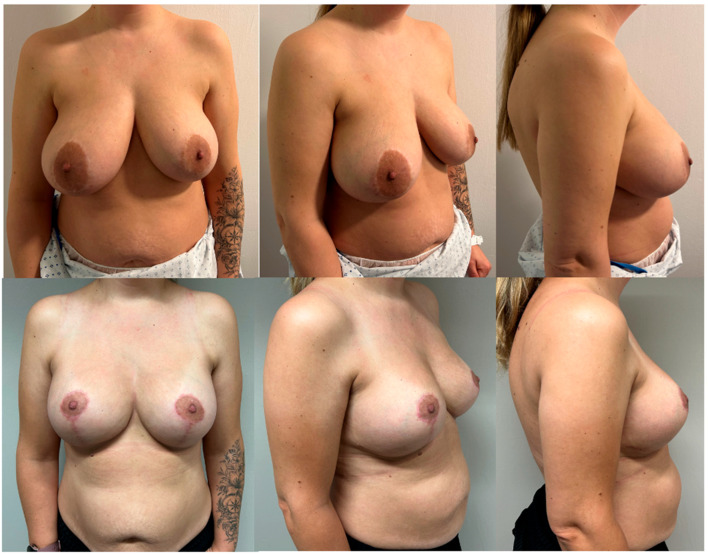
Pre-operative pictures (**above**) and post-operative pictures (**below**) of a 38-year-old patient with a grade 3 ptosis who underwent breast reduction and augmentation with 450 cc round high-profile implants. Results at 6-month follow-up.

**Table 1 medicina-60-00743-t001:** Patient characteristics.

	Mean ± SD	Range
Age (years)	29 ± 5	19–40
BMI	25.95 ± 2.95	22.2–32.5
Sternal notch-to-nipple distance (cm)		
- Right	28.43 ± 1.39	26–31
- Left	28.67 ± 1.25	27–31
Follow-up time (months)	14.17 ± 6.46	3–26
Implant size (cc)	352.39 ± 41.52	300–455
Amount of breast tissue removed (g)		
- Right	391.78 ± 86.28	312–657
- Left	392 ± 94.22	340–684

**Table 2 medicina-60-00743-t002:** Findings between pre-operative and post-operative questionnaires.

	Pre-Operative (n) *						Post-Operative (n) *						
	1	2	3	4	Mean	SD	1	2	3	4	Mean	SD	*p*-Value
How satisfied do you look in the mirror clothed?	8	7	8	0	2	0.83	0	1	5	17	3.79	0.55	<0.001 **
How satisfied that your breast size matches the rest of your body?	6	6	10	1	2.26	0.90	0	1	1	21	3.87	0.45	<0.001 **
How satisfied your bras fit?	5	12	6	0	2.04	0.69	0	1	2	20	3.83	0.48	<0.001 **
How satisfied are you with the cleavage you have when you wear a bra?	7	8	7	1	2.09	0.88	0	0	4	19	3.82	0.38	<0.001 **
How satisfied are you with the size of your breasts?	11	9	3	0	1.65	0.70	0	0	4	19	3.82	0.38	<0.001 **
How satisfied are you with how you look in the mirror unclothed?	19	1	3	0	1.30	0.68	1	2	2	18	3.61	0.82	<0.001 **

* The 4-point Likert scale converted numerically: 1 = very dissatisfied, 2 = somewhat dissatisfied, 3 = somewhat satisfied, 4 = very satisfied; ** statistically significant, *p* < 0.05.

## Data Availability

The raw data supporting the conclusions of this article will be made available by the authors on request.

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
