# Peer review of "Breast Reduction with Implants or Augmentation Reduction: Patient-Reported Outcomes from a Single-Centre Retrospective Cohort Analysis"

_medicina, 2024, doi:10.3390/medicina60050743_

Round 1
Reviewer 1 Report
Comments and Suggestions for Authors
Many thanks for asking me to review this study on a very controversial topic.
This is a retrospective case series of 25 patients undergoing breast reduction with symultaneous breast implants.
I have the following observations:
- In the discussion the authors "confidently say that breast reduction with implants has a similar complication profile to an augmentation mastopexy". I am not sure that the data reported allow for such a bold statement. It would be good to know how many augmentation-mastopexy have been performed by the senior author and what complication rate they had. I would then change the statement to reflect that complication are comparable, at least in their hands.
- Although I appreciate that it may be challenging to obtain longer follow-up data, I believe that checking patients' satisfaction 3-month post-operatively may not provide a reliable insight as it does not factor possible late complications and cosmetic sequelae. Why did you select the 3-month timepoint for post-operative PROMs?
- The PROMs questions are very similar to those included in the Breast-Q questionnaires, which are a validated tool. Why did you decide not to use them? This should be included in the limitations.
- An intraoperative picture or a diagram to clarify the implant pocket position would be very helpful
Author Response
Many thanks for asking me to review this study on a very controversial topic.
This is a retrospective case series of 25 patients undergoing breast reduction with symultaneous breast implants.
I have the following observations:
- In the discussion the authors "confidently say that breast reduction with implants has a similar complication profile to an augmentation mastopexy". I am not sure that the data reported allow for such a bold statement. It would be good to know how many augmentation-mastopexy have been performed by the senior author and what complication rate they had. I would then change the statement to reflect that complication are comparable, at least in their hands.
We thank the Reviewer for the comment. We amended the statement, accordingly, comparing augmentation mastopexy to already published complication profile data.
- Although I appreciate that it may be challenging to obtain longer follow-up data, I believe that checking patients' satisfaction 3-month post-operatively may not provide a reliable insight as it does not factor possible late complications and cosmetic sequelae. Why did you select the 3-month timepoint for post-operative PROMs?
3-month PROM is a standard for our practice and the UK regulator (CQC) encourages to have a solid system in place to capture patient’s outcome. Our practice encapsulates simple bilateral breast augmentation mammoplasty, breast reductions, augmentation mastopexy and breast reduction plus implants and body contouring. Unfortunately, we have found that most patients within the practice do not attend follow up past the 3-month mark. We do in fact send a PROMs questionnaire at 12-months, however PROMs at this 3-month allows us to record most of this patient cohort.
- The PROMs questions are very similar to those included in the Breast-Q questionnaires, which are a validated tool. Why did you decide not to use them? This should be included in the limitations.
We acknowledge that the practice’s questionnaire is very similar to BREAST-Q. The questionnaire was initially introduced within the practice as a standard questionnaire for all breast procedures in order to satisfy accreditation standards from the Care Quality Commission in the United Kingdom. It was not introduced with PROMs data collection and using a validated tool as the primary objective. We do acknowledge that this is a limitation for the study and we are in discussions with executive to change the pre-operative and post-operative questionnaire to use BREAST-Q moving forward. This will be mentioned as a limitation.
- An intraoperative picture or a diagram to clarify the implant pocket position would be very helpful
Thank you, a picture has been added to the manuscript.
Reviewer 2 Report
Comments and Suggestions for Authors
Appreciate authors attempt to present the outcome from a single institution. Main limitations of the study as acknowledged is as follows
1. Small cohort with very limited follow up. Even the mean follow up is 14.17 months, post operative PROM questionnaire was sent out at 3 months. So technically the PROM is at 3 months rather than 14.17 months. This needs to be clarified better in the discussion section.
2. Authors haven't discussed the option of auto augmentation and the advantage of implant assisted procedure over Auto augmentation using the reduction tissue.
3. The questionnaire is not designed to assess the functional benefits of alleviating back,shoulder and neck pain. So I am not sure conclusion is validated with the presented data.
4. Smaller number of patients with very limited follow up period does raise challenges in drawing conclusions on the results of this technique presented
Comments on the Quality of English Language
Throughout the manuscript, sentences are starting with some random numbers.
Author Response
Reviewer 2
Appreciate authors attempt to present the outcome from a single institution. Main limitations of the study as acknowledged is as follows
1. Small cohort with very limited follow up. Even the mean follow up is 14.17 months, post operative PROM questionnaire was sent out at 3 months. So technically the PROM is at 3 months rather than 14.17 months. This needs to be clarified better in the discussion section.
We thank the Reviewer for his comment. Yes, this has been further clarified in the discussion section
2. Authors haven't discussed the option of auto augmentation and the advantage of implant assisted procedure over Auto augmentation using the reduction tissue.
- Thank you for the comment. This has been added to the discussion.
3. The questionnaire is not designed to assess the functional benefits of alleviating back,shoulder and neck pain. So I am not sure conclusion is validated with the presented data.
- This has been reworded in the conclusion and discussion
4. Smaller number of patients with very limited follow up period does raise challenges in drawing conclusions on the results of this technique presented
- We acknowledge that a small patient population does present challenges in drawing conclusions. Our purpose is to add to the existing literature given that it is already quite sparse. With further studies, it may allow for a more robust systematic review or meta-analysis.
Reviewer 3 Report
Comments and Suggestions for Authors
Dear authors,
Congrats for your surgical work, but your manuscript did not bring evething new for literature.
Which is the novelty?
Only 17 references?
The discussion part should be more extensive.
Author Response
Reviewer 3
Dear authors,
Congrats for your surgical work, but your manuscript did not bring evething new for literature.
Which is the novelty?
We thank the Reviewer for his comment. The limitations of the study have been further elucidated in the discussion.
The novelty is that there are only a limited number of studies that describe this technique, which is part of the reason why there are not that many papers to cite. Our goal is to add to the body of existing literature and to demonstrate our practice’s results. Furthermore, as far as we know, there is no other study evaluating PROMs over time for breast augmentation reduction.
Only 17 references?
More references have been added to the manuscript.
The discussion part should be more extensive.
Thank you, we extended the discussion, taking into account all the Reviewers’ comments.
Round 2
Reviewer 1 Report
Comments and Suggestions for Authors
I am happy with the changes made
Reviewer 3 Report
Comments and Suggestions for Authors
Now it is better.